# SHMT2 reduces fatty liver but is necessary for liver inflammation and fibrosis in mice
Guohua Chen[1], Guoli Zhou[2], Lidong Zhai[3], Xun Bao[4], Nivedita Tiwari[5], Jing Li[4], Emilio Mottillo [5,6] & Jian Wang [1] ✉

Non-alcoholic fatty liver disease is associated with an irregular serine metabolism. Serine hydroxymethyltransferase 2 (SHMT2) is a liver enzyme that breaks down serine into glycine and one-carbon (1C) units critical for liver methylation reactions and overall health. However, the contribution of SHMT2 to hepatic 1C homeostasis and biological functions has yet to be defined in genetically modified animal models. We created a mouse strain with targeted *SHMT2* knockout in hepatocytes to investigate this. The absence of *SHMT2* increased serine and glycine levels in circulation, decreased liver methylation potential, and increased susceptibility to fatty liver disease. Interestingly, *SHMT2*-deficient mice developed simultaneous fatty liver, but when fed a diet high in fat, fructose, and cholesterol, they had significantly less inflammation and fibrosis. This study highlights the critical role of SHMT2 in maintaining hepatic 1C homeostasis and its stage-specific functions in the pathogenesis of NAFLD.

Non-alcoholic fatty liver disease (NAFLD) is a significant health issue that affects around a quarter of the global population. This condition is progressive and includes hepatic steatosis, steatohepatitis, liver fibrosis/cirrhosis, and hepatocellular carcinoma, which can be life-threatening. Despite the high prevalence of NAFLD, no approved treatments are available, highlighting the urgency for more research and development in this field[1,2].

NAFLD can arise from impaired one-carbon (1C) metabolism, a metabolic process that involves the transfer of bioactive methyl groups (1C units) via pathways such as the folate and methionine cycles[3,4]. These methyl groups play a vital role in producing essential metabolic intermediates, including nucleotides and phospholipids, and participate in gene regulation[5]. Recent research on patients and mice has demonstrated that maintaining proper hepatic methylation potential, as determined by the concentration of the main 1C metabolite S-adenosylmethionine (SAM), is crucial for preventing NAFLD. SAM-dependent phosphatidylcholine (PC) synthesis is pivotal for upholding the integrity of the phospholipid metabolome. This, in turn, either supports the assembly of very low-density lipoprotein (VLDL), which promotes hepatic triglyceride (TG) disposal, or prevents endoplasmic reticulum (ER) stress, thereby reducing the risk of hepatocyte de novo lipogenesis (DNL) hyperactivation. Ultimately, this

process helps reduce the risk of hepatic fat accumulation and NAFLD progression[6–9].

Serine is the primary supplier of 1C units in the human body, and isotopic tracing has shown that almost all methyl groups used to replenish the SAM 1C pool are derived from this amino acid[10]. Studies indicate that individuals with NAFLD and metabolic syndrome (MetS) tend to have lower levels of serine and glycine in circulation[11,12]. These two amino acids have been suggested as promising biomarkers for diagnosing NAFLD and guiding nutritional interventions[12–15]. The enzyme serine hydroxymethyltransferase (SHMT) breaks down serine into glycine, producing 1C for methylating folate in the folate cycle[16]. The methylated folate species supports the methionine cycle, which produces SAM. Systems biology modeling suggests that NAFLD may be caused by a deficiency in serine and reduced SHMT activity[17].

Mammals have two SHMT paralogs: cytosolic SHMT1 and mitochondrial SHMT2[18]. While the global *SHMT1* knockout mouse model has been observed to be healthy, fertile, and viable with elevated hepatic SAM levels, indicating that *SHMT1* may not be necessary to maintain the 1C pool[19], the whole-body *SHMT2* deletion leads to embryonic lethality, indicating its crucial role[20,21]. However, global *SHMT2* knockout is unsuitable for simulating metabolic diseases that usually impact adults. It is yet to

[1]Department of Pathology, Wayne State University School of Medicine, Detroit, MI 48202, USA. [2]Biomedical Research Informatics Core, Clinical and Translational Sciences Institute, Michigan State University, East Lansing, MI 48824, USA. [3]Department of Pathology, University of Michigan, Ann Arbor, MI 48109, USA. [4]Department of Oncology, Wayne State University School of Medicine, Detroit, MI 48202, USA. [5]Hypertension and Vascular Research Division, Henry Ford Hospital, Detroit, MI 48202, USA. [6]Department of Physiology, Wayne State University School of Medicine, Detroit, MI 48202, USA. ✉e-mail: jianwang@med.wayne.edu

be defined how the critical serine-catabolic enzyme SHMT2 regulates 1C homeostasis and contributes to metabolic health in adult animals.

We have modeled hepatic *SHMT2* deletion in mice, which survives embryonic development. Our research has revealed the crucial role of *SHMT2* in maintaining hepatic methylation potential in adult animals. We have further discovered that SHMT2 has a nonlinear function in NAFLD: inhibiting hepatic steatosis but supporting liver fibrosis development. These results have established the pivotal role of SHMT2 in regulating hepatic 1C levels while uncovering its potential therapeutic values against NAFLD at various stages.

## Results

### SHMT2 is essential for maintaining hepatic methylation potential

In mice, whole-body deletion of *SHMT2* causes embryonic lethality due to faulty fetal hematopoiesis in embryonic liver[22], making it impossible to assess adult phenotypes. To address this, we created a floxed *SHMT2* mouse allele (SHMT2$^{fl/fl}$) with two loxP sites spanning exons 3–12, allowing for conditional targeting by lox/Cre recombination (Fig. S1a). Next, we crossbred this allele with the Albumin-Cre allele[23] to specifically target the hepatocytes in the liver. Successful recombination at the *SHMT2* locus was confirmed by PCR-based mouse genotyping (Fig. S1b). The liver is a vital metabolic organ that expresses a significant amount of SHMT2[24]. The resulting mouse strain has been named SHMT2$^{HKO}$. Mice possessing homozygous SHMT2$^{HKO}$ genes exhibited no discernible abnormalities at birth and were capable of reproduction. Compared to the SHMT2$^{fl/fl}$ control, their liver SHMT2 protein levels decreased by approximately 50% at birth,

continued to decrease to about 30% at weaning, and were almost undetectable in adulthood (Fig. 1a). Throughout the process of liver development, there is a notable increase in the number of hepatocytes while the number of non-parenchymal cells (NPCs) gradually decreases after birth[25]. This observation suggests that targeting hepatocyte *SHMT2* in SHMT2$^{HKO}$ mice was successful, without affecting SHMT2 expression in NPCs.

We examined the role of the SHMT2 enzyme in the breakdown of the serine amino acid, which results in the production of glycine. To do this, we fed SHMT2$^{HKO}$ mice a chow diet until they were 22 weeks old and then analyzed their circulating amino acid profiles. Our findings showed significant changes in the levels of serine, glycine, and aspartate in the SHMT2$^{HKO}$ mice (Fig. 1b–e). We found that the absence of *SHMT2* increased the levels of serine in the bloodstream, indicating a decrease in hepatic serine catabolic activity. Interestingly, the SHMT2$^{HKO}$ mice had three times more serum glycine levels than the control mice, which cannot solely be attributed to the expected decrease in glycine production. Additionally, deleting hepatic *SHMT2* led to lowered serine levels and elevated glycine levels in the liver (Fig. 2a, b). These findings suggest that deleting hepatic *SHMT2* has metabolic impacts beyond the breakdown of serine and the production of glycine in intact animals.

We conducted targeted metabolomics analysis of liver metabolites to evaluate the metabolic reprogramming that occurs with the deletion of hepatic *SHMT2*. We identified 48 metabolites that were differentially expressed (DEM) out of 188 metabolites analyzed, with a cutoff of $p < 0.05$ and FDR < 0.20 (Fig. 2c, Supplementary Data 1, 2). Pathway enrichment analysis showed that the DEMs were mainly localized within the top

**Fig. 1 | The deletion of *SHMT2* in the liver significantly elevates the levels of glycine in circulation. a** SHMT2 protein levels were tested in the *SHMT2*-deficient (HKO) and control (fl/fl) mice liver at specific times using Western blotting. **b** The heatmap shows amino acid expression in control and *SHMT2*-deficient mice. **c–e** Bar graphs illustrate the variation in circulating amino acid levels between *SHMT2*-deficient and control mice. (Mean ± SD; *$p < 0.05$, **$p < 0.01$, ****$p < 0.0001$; *t*-test).

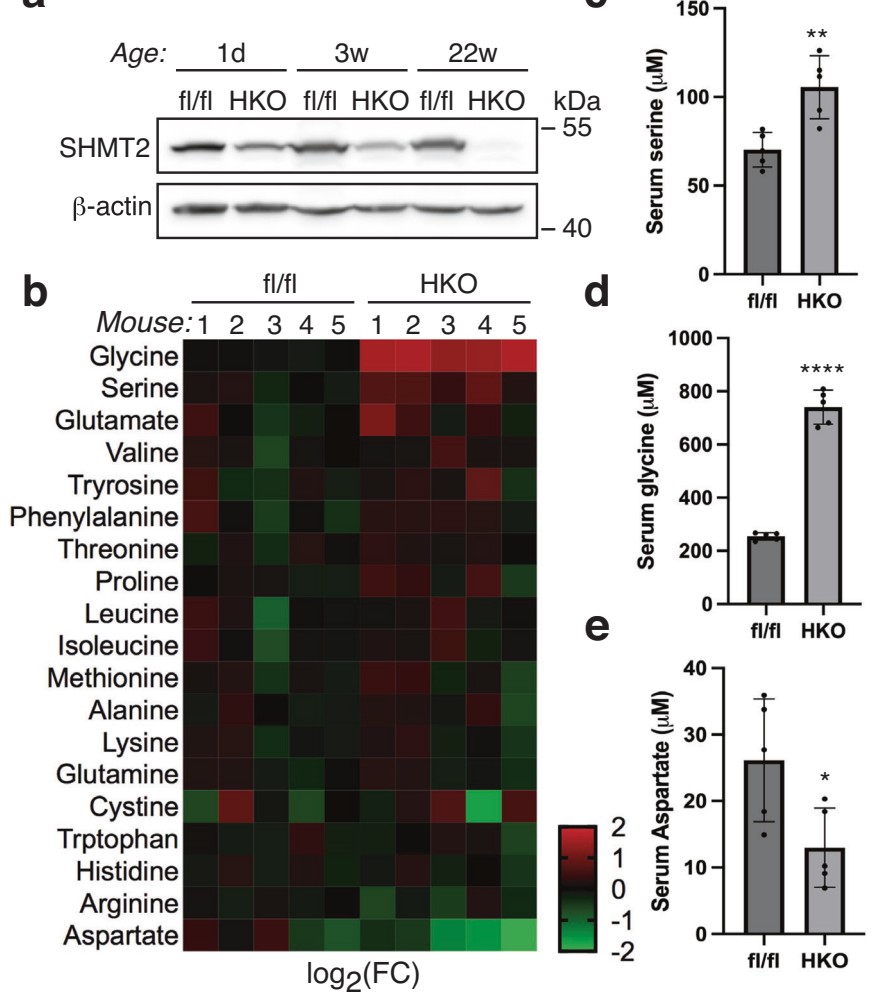

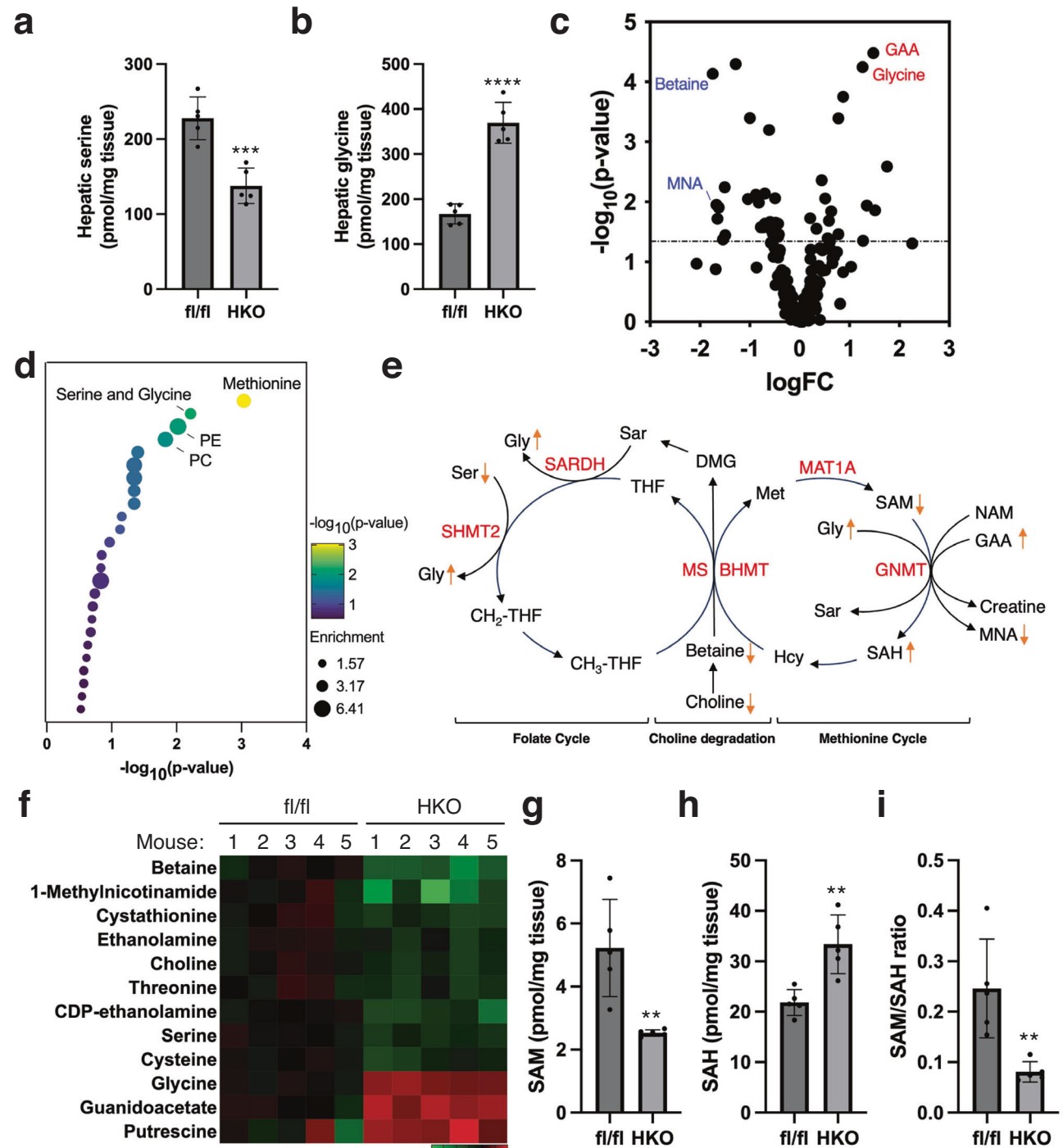

**Fig. 2 | *SHMT2* deficiency reduces liver methylation potential. a, b** Bar graphs illustrate serine and glycine levels in the livers of *SHMT2*-deficient (HKO) and control (fl/fl) mice. (Mean ± SD; ***$p < 0.001$, ****$p < 0.0001$; *t*-test). **c** Comparison of metabolite levels in mouse livers (HKO vs. fl/fl) shown on a volcano plot. **d** Bubble plot showing enriched pathways of differentially expressed metabolites in mouse livers (HKO vs. fl/fl). **e** A cartoon diagram showing the mapping of 1C DEMs in the folate, methionine cycles, and choline degradation pathways. Ser serine, Gly glycine, Sar sarcosine, DMG dimethylglycine, THF tetrahydrofolate, $CH_2$-THF methylene tetrahydrofolate, $CH_3$-THF methyl tetrahydrofolate, Met methionine, Hcy

homocysteine, SAM S-adenosylmethionine, SAH S-adenosylhomocysteine, NAM nicotinamide, MNA 1-methylnicotinamide, GAA guanidinoacetate, SHMT2 serine hydroxymethyltransferase 2, SARDH sarcosine dehydrogenase, MS methionine synthase, BHMT betaine homocysteine methyltransferase, MAT1A methionine adenosyltransferase 1A, GNMT glycine N-methyltransferase. **f** The heatmap shows the DEM levels of the methionine, serine and glycine, PE and PC pathways in mouse livers. **g–i** Liver SAM and SAH levels were analyzed to assess hepatic methylation potential in HKO and fl/fl mice. (Mean ± SD; **$p < 0.01$; *t*-test).

pathways of the methionine metabolism, serine and glycine metabolism, phosphatidylethanolamine (PE) biosynthesis, and phosphatidylcholine (PC) biosynthesis, which are crucial components of the 1C metabolic process and the related biogenesis of phospholipids (Fig. 2d). In the liver of SHMT2[HKO] mice, we observed a distinct pattern of metabolite expression, where the levels of 1C-donating metabolites, including serine, choline, betaine, and 1-methyl nicotinamide (1-MNA), decreased, and the levels of 1C-recipient metabolites, including glycine, guanidinoacetate (GAA), and putrescine, increased (Fig. 2e, f). These findings suggest that the deletion of *SHMT2* has led to a hepatic 1C deficiency. We also measured the levels of SAM and S-adenosyl homocysteine (SAH), which indicates methylation potential[26]. We found that SHMT2[HKO] mice had decreased SAM levels and increased SAH levels in the liver, resulting in a significant decline in the SAM/SAH ratio (Fig. 2g–i). These results highlight the critical role of *SHMT2* in maintaining the liver's 1C pool and methylation potential, even if other pathways can also supply 1C units in hepatocytes (Refer to pathway diagram in Fig. 2e).

Glycine plays a crucial role in the formation of glutathione, which is the primary antioxidant of the body. We observed that SHMT2[HKO] mice showed a significant increase in glycine levels in their liver and bloodstream. This led us to investigate whether it affected the glutathione concentration in the liver. However, we found no significant change in hepatic glutathione abundance due to *SHMT2* deletion (Fig. S2).

### The absence of *SHMT2* leads to the development of simultaneous fatty liver
As reduced liver methylation capacity increases susceptibility to NAFLD, we examined if lack of *SHMT2* impacts liver pathogenesis. We observed SHMT2[HKO] mice under chow-diet-feeding conditions for up to 22 weeks. After analyzing their growth kinetics, we found that the absence of *SHMT2* did not cause significant changes in body weight or body composition (Fig. 3a, b). *SHMT2* knockout animals displayed no differences in glycemia, glucose tolerance, insulin sensitivity, circulating triglyceride (TG), or total cholesterol (TC) levels compared to the control group (Fig. 3c–e). On the other hand, the deletion of *SHMT2* significantly increased hepatic levels of TG and TC and circulating alanine aminotransaminase (ALT) levels (Fig. 3f, g). Histological analysis and Oil Red O staining consistently showed that the abundance and size of liver lipid droplets significantly increased with *SHMT2* deletion (Fig. 3h). Our findings suggest that *SHMT2* prevents hepatic steatosis and protects the liver from steatosis damage in mice under chow-fed conditions.

### The absence of *SHMT2* worsens hepatic steatosis caused by an over-nutritious diet, but it lessens inflammation and fibrosis in the liver
Overnutrition is known to exacerbate NAFLD. To investigate the effects of NAFLD progression in rodents, the AMLN diet, which is high in fat, fructose, and cholesterol, is commonly utilized[27,28]. In this study, SHMT2[HKO] mice were switched from their regular diet to the AMLN diet at four weeks of age and were harvested at 22 weeks of age, following 18 weeks on the AMLN diet. The SHMT2[HKO] mice on the AMLN diet demonstrated increased fat mass, resulting in higher body weight compared to the control group (Fig. 4a). Biochemical analyses revealed that these mice had elevated levels of circulating and hepatic TG and TC (Fig. 4b, c). Additionally, they exhibited increased fasting glycemia and more severe glucose disposal impairment and insulin insensitivity after 14 and 16 weeks of exposure to the AMLN diet (Fig. 4d, e). Liver histological analysis indicated that SHMT2[HKO] mice had increased lipid droplet deposition in hepatocytes, mainly exhibiting excessive microvesicular steatosis in the pericentral zone, resulting in higher steatosis grading compared to the control (Fig. 4f, g). However, the SHMT2[HKO] mice demonstrated significantly reduced lobular inflammation and pericellular fibrosis, leading to a significant decrease in inflammation and fibrosis pathological grading compared to the control group (Fig. 4f, g). The SHMT2[HKO] mice exhibited a notable increase in the levels of circulating ALT (Fig. 4h). We concluded that *SHMT2* deletion in hepatocytes worsens liver steatosis and injury but reduces inflammation and fibrosis in the AMLN diet treatment.

### The deletion of *SHMT2* does not have a discernible impact on the expression of hepatic electron transport chain proteins
The proper oxidation of mitochondria is crucial in disposing of fatty acids and preventing NAFLD. Previous research has indicated that SHMT2 is critical in maintaining electron transport chain (ETC) protein expression and mitochondrial respiration in murine embryonic fibroblasts and human HEK293 cells[22,29–31]. However, upon studying SHMT2[HKO] mice livers under chow-fed conditions, Western blotting analysis revealed that the protein markers for each respiratory complex were expressed at similar levels in comparison to control mice (Fig. 5a). We isolated primary hepatocytes from mice. Western blot analysis confirmed the complete absence of SHMT2 protein in hepatocytes from SHMT2[HKO] mice (Fig. 5b, top). Furthermore, Seahorse respirometry analysis conducted on the primary mouse hepatocytes demonstrated that the *SHMT2* knockout had minimal impact on basal and maximal cellular oxygen consumption (Fig. 5b, bottom). After the *SHMT2* gene was deleted, the AML12 mouse hepatocyte cell line produced similar results (Fig. S3). Therefore, it can be concluded that SHMT2 is not essential for the mitochondrial respiration of mouse hepatocytes.

### Transcriptomics profiling shows that deleting *SHMT2* promotes the expression of the DNL pathway while reducing inflammation and fibrosis pathways in the diet-induced NAFLD model
Using RNAseq, we analyzed the liver transcriptome to explore how *SHMT2* affects the progress of diet-induced NAFLD in mice. We discovered a list of genes with varying expression levels between SHMT2[HKO] and control mice when fed an AMLN diet (Supplementary Data 3, 4). Gene set enrichment analysis (GSEA) showed that upregulated genes were highly enriched in lipogenesis pathways, while downregulated genes were highly enriched in the inflammatory response and extracellular matrix remodeling pathways (Fig. 6a–c). Our qRT-PCR analyses revealed that the AMLN diet decreased the hepatic expression of de novo lipogenesis (DNL) genes in SHMT2[fl/fl] control mice, except for *Scd1*. However, this decrease was not present in SHMT2[HKO] mice. Furthermore, SHMT2[fl/fl] mice exhibited elevated hepatic *Scd1* expression responsive to the AMLN diet, which was even more pronounced in SHMT2[HKO] mice (Fig. 6d). Western blotting confirmed the upregulated hepatic protein expression of DNL genes in SHMT2[HKO] mice under AMLN diet feeding (Fig. 6e). Conversely, the AMLN diet significantly increased the expression of hepatic inflammatory and fibrotic genes in SHMT2[fl/fl] mice, but this was significantly reduced by hepatic *SHMT2* deletion (Fig. 6f, g). These findings suggest that SHMT2 plays a crucial role in suppressing genes involved in de novo lipogenesis, while promoting inflammation and fibrosis pathways in a diet-induced model of NAFLD.

### The relationship between *SHMT2* genetic variations and expression with metabolic diseases in humans
We conducted research on functional genomic data using the Common Metabolic Disease Knowledge Portal (CMDKP)[32] to investigate potential correlations between single nucleotide polymorphisms (SNP) in the *SHMT2* gene and metabolic phenotypes contributing to metabolic diseases. Our analysis was focused on specific SNPs that met the following criteria: they are located between the start and stop sites of the *SHMT2* gene's transcription, present in the 1000 Genomes reference database, and part of the "bottom-line" meta-analyzed genetic associations for the relevant phenotype in CMDKP database. These variations are particularly suitable for identifying the linkage disequilibrium of the *SHMT2* gene, which plays a role in developing complex metabolic diseases. Our results indicate that SNPs in *SHMT2* are significantly associated with changes in metabolic parameters such as circulating lipids and lipoproteins, body mass indexes, and diabetic indicators (Fig. 7a and Table S1). These findings suggest that genetic variations in *SHMT2* may influence *SHMT2* gene expression and activity,

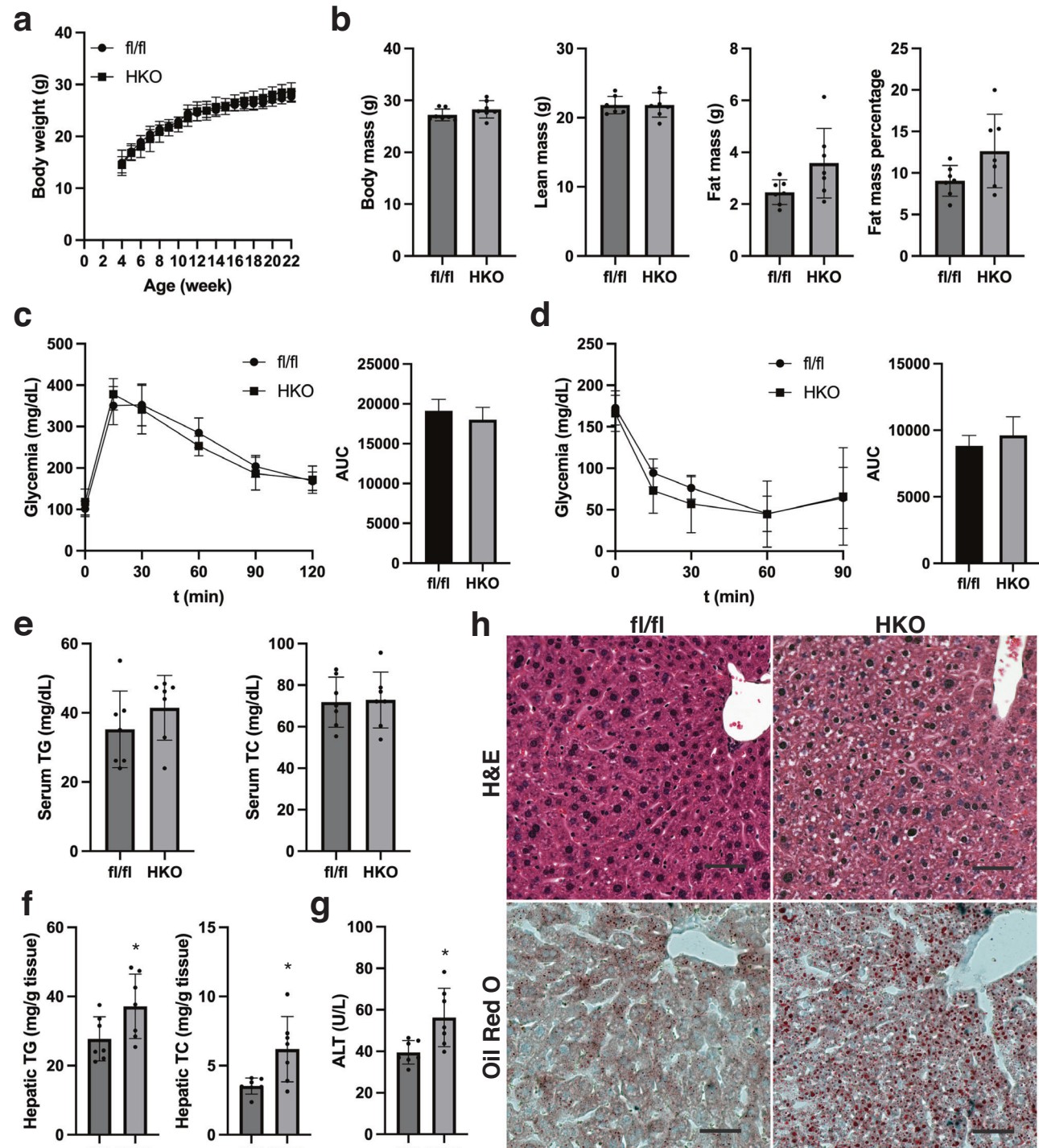

**Fig. 3 | The development of hepatic steatosis occurs concomitantly with the deficiency of SHMT2. a** Growth kinetics of *SHMT2*-deficient (HKO) and control (fl/fl) mice. **b** Body composition of 22-week-old mice was measured using Eco-MRI during harvesting. **c** The glucose intolerance test was conducted when the subject reached 18 weeks of age (*n* = 7). **d** The insulin sensitivity test was conducted when the subject reached 20 weeks of age (*n* = 6). **e** Measuring the levels of total tria-cylglycerol (TG) and total cholesterol (TC) in circulation. **f** Levels of TG and TC in the liver. (Mean ± SD; *\*p* < 0.05; *t*-test). **g** Levels of ALT in circulation (Mean ± SD; *\*p* < 0.05; *t*-test). **h** The mouse liver sections were stained using H&E (top) and Oil Red O (bottom). Scale bar: 100 μm.

ultimately impacting the development and outcome of metabolic syndrome in humans. Upon conducting a more in-depth analysis of *SHMT1*, which encodes the cytosolic counterpart of SHMT2, it was observed that the *SHMT1* gene SNPs are also significantly associated with human metabolic phenotypes that define metabolic syndrome (Fig. S4 and Table S2). These results indicate that both *SHMT1* and *2* are associated with susceptibility to metabolic syndrome in human populations.

Furthermore, we analyzed GSE datasets to determine whether *SHMT2* expression is altered in liver specimens of NAFLD patients. Our meta-analysis demonstrated that *SHMT2* mRNA levels are significantly higher in NASH patients compared to healthy controls in two data sets (Fig. 7b). This indicates that *SHMT2* is a metabolic gene that significantly associates with the development and outcome of metabolic syndrome and liver disease in human populations.

**Fig. 4 | SHMT2 deficiency worsens fatty liver but improves AMLN diet response by reducing inflammation and fibrosis. a** Eco-MRI measured HKO and fl/fl mice's body composition after 18 weeks on AMLN diet at 22 weeks old (Mean ± SD; *$p < 0.05$, **$p < 0.01$; $t$-test). **b** The levels of total triacylglycerol (TG) and total cholesterol (TC) in mouse circulation after 18 weeks on AMLN diet. (Mean ± SD; *$p < 0.05$; $t$-test). **c** The levels of TG and TC in mouse livers after 18 weeks on AMLN diet. (Mean ± SD; *$p < 0.05$; $t$-test). **d** Glucose tolerance test conducted on mice after 14 weeks on AMLN diet (Mean ± SD; $n = 6$; ***$p < 0.001$; $t$-test). **e** Insulin sensitivity test conducted on mice after 16 weeks on AMLN diet. (Mean ± SD; $n = 6$; *$p < 0.5$; $t$-test). **f** H&E (left), Oil Red O (middle), and Sirius Red (Right) staining of mouse liver sections after 18 weeks on AMLN diet. Scale bar: 100 μm. **g** Pathological grading of liver steatosis (left), lobular inflammation (middle), and fibrosis (right). (Mean ± SD; *$p < 0.05$, **$p < 0.01$; $t$-test). **h** Levels of ALT in circulation (Mean ± SD; **$p < 0.01$; $t$-test).

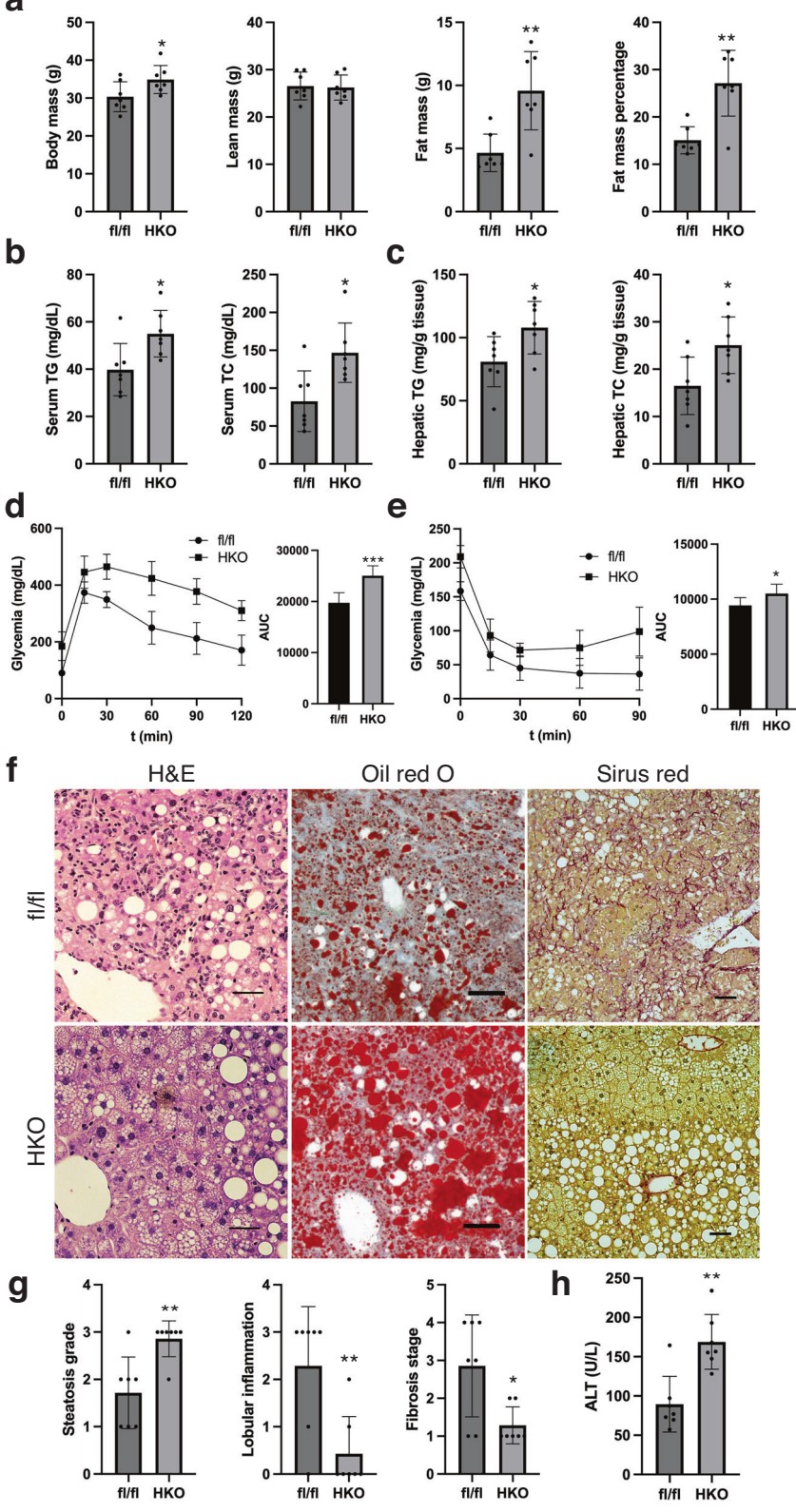

## Discussion

The liver plays a crucial role in metabolic processes that require a significant amount of 1C units, responsible for over 80% of the body's methylation reactions[33]. To acquire the necessary 1C pool, two major pathways are utilized: the liver-specific choline/betaine degradation pathway and the universal serine catabolic pathway. A vital question arises on whether these pathways are interchangeable. Past research has indicated that disrupting the critical 1C-donating enzyme betaine homocysteine methyltransferase (BHMT) in the choline/betaine degradation pathway reduces hepatic methylation potential in mice[34]. Our latest findings indicate that disrupting serine catabolic enzyme SHMT2 in the liver reduces SAM while increasing SAH, resulting in a diminished SAM/SAH ratio and indicating reduced

**Fig. 5 | The deletion of *SHMT2* does not impact the expression of ETC proteins in hepatocytes.**
**a** Western blotting measures each ETC complex's protein levels in the livers of *SHMT2*-deficient (HKO) and control (fl/fl) mice. The individual complex markers were NDUFS1 (complex I), SDHB (complex II), UQCRC2 (complex III), COX II (complex IV), and ATP5A (complex V). **b** Top panel: Western blotting confirms *SHMT2* knockout in primary hepatocytes. Bottom panel: respirometry analysis for the *SHMT2* knockout (KO) and wild-type (WT) primary hepatocytes (*n* = 11).

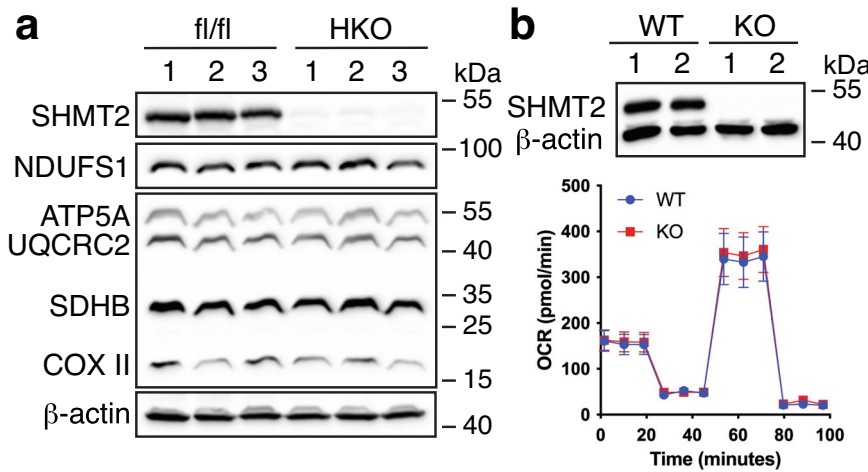

**Fig. 6 | Transcriptomics profiling reveals that *SHMT2* deficiency increases DNL gene expression while reducing inflammatory and fibrotic gene expression in mouse livers. a** RNA-seq analysis showed pathway enrichment of differentially expressed genes (DEGs) in the livers of both *SHMT2*-deficient and control mice after being on the AMLN diet for 18 weeks. **b** GSEA plots show DEG enrichment for fatty acid biosynthesis and inflammatory response pathways in the liver of HKO and fl/fl mice fed the AMLN diet. **c** RNA-seq detected DEGs of fatty acid biosynthesis (left) and inflammatory response (right) the AMLN-fed mouse livers shown in heat maps.

**d** The gene expression of the DNL pathway was measured in the livers of specified mice using qRT-PCR (Mean ± SD; *p < 0.05, **p < 0.01, ***p < 0.001, ****p < 0.0001; *t*-test). **e** The protein expression of DNL enzymes in mouse livers fed with the AMLN diet was measured by Western blotting. The gene expression of the inflammatory (**f**) and fibrotic (**g**) pathways was measured in the livers of specified mice using qRT-PCR. (Mean ± SD; *p < 0.05, **p < 0.01, ***p < 0.001, ****p < 0.0001; *t*-test).

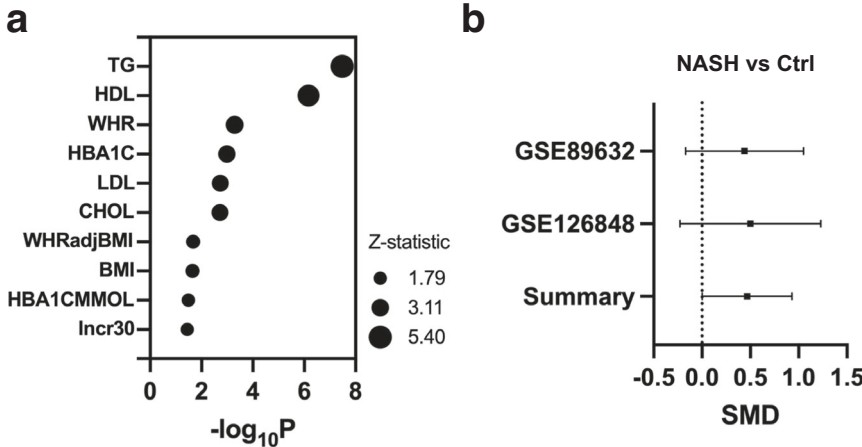

**Fig. 7 | Metabolic syndrome and non-alcoholic fatty liver disease (NAFLD) in humans are linked to genetic and expression level variations of SHMT2. a** SHMT2 polymorphism affects metabolic syndrome parameters in humans, as shown in the bubble plot. **b** SHMT2 expression is upregulated in NASH patients' liver samples, per the meta-analysis shown in the forest plot. The Y axis displays GEO accession numbers while the X axis represents the standard mean difference (SMD) of SHMT2 gene expression between NASH and healthy control (Ctrl) groups.

methylation potential. Moreover, SHMT2 loss significantly impacts methionine metabolism and the methyl metabolome, leading to a 1C deficiency in the liver. The methyl donors or carriers, such as serine, choline, betaine, sarcosine, and 1-MNA, decrease, while the methyl recipients, such as glycine and GAA, increase. Our results suggest that both serine catabolism and choline degradation are essential to fulfill the liver's transmethylation demands for 1C units.

An unexpected discovery was made when the enzyme SHMT2, responsible for producing glycine, was removed from hepatocytes. Surprisingly, the SHMT2 removal resulted in an increase in glycine levels both in the liver and bloodstream. These findings may reflect the critical role of glycine in regulating the 1C balance in the liver through participating in a regulatory process called "methyl sink." When there is an excess of 1C, glycine is converted into sarcosine by GNMT, which consumes SAM. Conversely, when there is a shortage of 1C, sarcosine dehydrogenase (SARDH) breaks down sarcosine into glycine, producing a unit of 1C with the help of folate (Refer to pathway diagram in Fig. 2e)[35]. SAM regulates the activation of GNMT through allosteric regulation, and it is known that GNMT is the most abundant liver methyltransferase, accounting for 1% of liver soluble protein[36]. The increase in glycine levels in mice without hepatic SHMT2 could be due to SAM depletion, leading to the deactivation of GNMT and activation of SARDH. This, in turn, inhibits glycine methylation and mobilizes sarcosine storage to balance the 1C pool deficiency, producing a significant amount of glycine in the liver that can affect circulation. Our research suggests that glycine may be a central component of hepatic regulatory "methyl sink", playing a vital role in balancing the 1C pool of the liver.

Deleting SHMT2 in whole mice arrested erythroblast development in the fetal liver, ultimately leading to embryonic death at E13.5[22]. This highlights the critical role of serine-derived 1C units in facilitating the folate cycle and nucleotide synthesis, which are crucial for cell proliferation. Interestingly, mice with SHMT2 deletion specific to their hepatocytes were able to survive without any developmental abnormalities. This suggests that hepatocytes have an alternative 1C source for the folate cycle, such as sarcosine, which is derived from the degradation of choline and betaine. It is possible that the liver has a unique pathway for breaking down choline/betaine/sarcosine, which can compensate for the loss of SHMT2 in hepatocytes. Our research sheds light on how hepatocytes can adapt to acquire 1C units to support the folate cycle and cell proliferation, ultimately promoting survival.

This study has uncovered a novel function of SHMT2 in inhibiting fatty liver development. While previous studies had connected changes in serine metabolism to NAFLD[12,17], our findings offer mouse genetic evidence demonstrating the significance of SHMT2-mediated serine catabolism in the development of NAFLD. Previous research has shown that SHMT2 plays a crucial role in maintaining the electron transport chain (ETC) and mitochondrial respiration in mouse embryonic fibroblasts and HEK293

cells[22,29–31], which could affect lipid disposal through mitochondrial fatty acid oxidation. Interestingly, disrupting SHMT2 did not impact cellular oxygen consumption or ETC protein expression in liver cells, indicating that liver cells have a unique mechanism to safeguard mitochondria, akin to serine catabolism. It is possible that the 1C units produced by the liver-specific choline degradation pathway may be enough to replenish the mitochondrial 1C-folate pool, but not enough to compensate for SAM deficiency in SHMT2-deficient hepatocytes. SAM dysregulation can, in turn, lead to an accumulation of fat in the liver.

Previous studies revealed that the restriction of glycine availability had a detrimental impact on glutathione biosynthesis and fatty acid oxidation, leading to the exacerbation of hepatic steatosis and fibrosis in mice[14]. Despite the significant increase in circulating and hepatic glycine levels, SHMT2[HKO] mice did not exhibit notable changes in hepatic glutathione abundance or hepatocyte respiratory capacity. These findings suggest that the disruption of 1C homeostasis, rather than glycine availability, may be the underlying cause of hepatic pathogenesis in SHMT2[HKO] mice.

Depleting BHMT or methionine adenosyltransferase 1 A (MAT1A) in mice lowers methylation potential, resulting in NAFLD[6,34]. This is due to the impaired SAM-dependent synthesis of phosphatidylcholine (PC), which causes defective very-low-density lipoprotein (VLDL) assembly in the liver and, thus, fat buildup. As deleting SHMT2 reduces SAM levels and disrupts liver methylation potential, further research is necessary to confirm whether SHMT2-mediated 1 C synthesis is necessary to maintain liver PC synthesis, VLDL assembly, and fat disposal.

Our research on the diet-induced development of non-alcoholic steatohepatitis (NASH) revealed that SHMT2 disruption can have both positive and negative effects on liver health. While it worsened steatosis, it could also aid in reducing liver inflammation and fibrosis in cases with an inflammatory and fibrogenic diet. SHMT2 disruption increased the hepatic DNL pathway due to the derepression of DNL genes when there was an overload of lipids. Sterol regulatory element binding proteins (SREBPs) recognize nutrient signals and mature in the ER to activate lipid synthesis genes[37]. It is critical to maintain the integrity of the ER membrane, which requires adequate PC content to prevent excessive SREBP activation[38]. Thus, the increase in liver DNL genes may be due to lower SAM-dependent PC synthesis, decreased ER PC content, and subsequent SREBP derepression in SHMT2 deficient mice. Further research is necessary to determine whether SHMT2 disruption may worsen fatty liver under AMLN diet treatment by decreasing VLDL-mediated hepatic lipid disposal and increasing SREBP-mediated hepatic lipid synthesis.

A significant discovery of this study indicates that SHMT2 is necessary for developing liver inflammation and fibrosis. Despite having a significant increase in liver steatosis and injury, mice lacking SHMT2, when given the AMLN diet, had reduced collagen deposition and immune cell infiltration in their livers, as revealed in the histological analyses, as well as lower levels of

inflammation and fibrosis-related genes. The SHMT2[HKO] mice showed increased lipogenesis that contributed directly to liver steatosis and injury. However, this increase was not compatible with reduced inflammation and fibrogenesis, which are typically stimulated by liver steatosis and injury. It is possible that additional *SHMT2* activity is required to activate the liver's immune and fibrogenic response to steatosis and injury. Liver injury triggers inflammation and fibrosis by instigating interaction between hepatocytes, immune cells, and fibrogenic cells[39]. The animal model in the current study specifically targeted *SHMT2* in the hepatocytes of the mouse liver, highlighting the importance of communication between these cells. This communication may involve serine catabolic activity derived from hepatocytes, which supports the activation and expansion of immune and fibrogenic cells in the liver, likely by generating methyl folate species and SAM. These metabolites are crucial for nucleotide synthesis and transmethylation reactions, which support cell proliferation and epigenetic gene regulation. Further research is necessary to determine if the serine-derived 1 C activities from hepatocytes regulate the metabolism or signaling of immune and fibrogenic cells in supporting their proliferation and functional expansion.

A significant association was found between the *SHMT2* genetic variation, its expression, and increased susceptibility to NAFLD and MetS in human populations, as well as SHMT2's stage-specific regulation of the onset and progression of NAFLD in the model animals. These findings suggest that precision medicine strategies are necessary when considering therapeutic interventions for serine catabolism in NAFLD at different stages, despite its significant value as a therapeutic target for the disease. It is also important to be cautious about interfering with SHMT2 during pregnancy because it is essential for embryonic development.

## Methods
### Animal studies
All studies conducted were authorized by the Institutional Animal Care and Use Committee of Wayne State University. We have complied with all relevant ethical regulations for animal use. Mice were maintained under a 12-hour day-night cycle and fed ad libitum with a control chow diet (#D12450K, Research Diets) or an AMLN diet (#D17010103, Research Diets). The SHMT2[fl/fl] strain was generated in the C57BL/6 J background by flanking exon 3–12 of the SHMT2 gene with loxP sites using CRISPR/Cas9-mediated genome editing (Cyagen). The Albumin-Cre strain in the C57BL/6 J background was obtained from the Jackson Laboratory (Strain# 003574). The SHMT2[HKO] strain was generated by crossbreeding the SHMT2[fl/fl] with the Albumin-Cre strain.

In the standard experiment protocol, male mice were weaned at three weeks of age and began consuming a chow diet. The mice were either kept on a chow diet or switched to the AMLN diet at 4 weeks of age. They were then exposed to the AMLN diet for 18 more weeks until they were 22 weeks old. GTT and ITT analyses were performed at 18 and 20 weeks of age, respectively. Body composition analysis was performed at 22 weeks of age on an EchoMRI instrument. For tissue harvesting, mice were fasted for 5 h. Blood samples were drawn from the inferior vena cava of the anesthetized animals for biochemical analysis. Dissected liver tissues were either flash-frozen in liquid nitrogen for biochemical analysis or fixed in 10% neutral buffered formalin for histological analysis.

### Mouse primary hepatocyte isolation and cell culture
To obtain primary mouse hepatocytes, a two-step collagenase perfusion method was used. The procedure involved opening the abdominal cavity and cannulating the inferior vena cava (IVC) with a catheter. The portal vein was incised to allow sufficient outflow during perfusion. The liver was perfused with buffer A (calcium-free HBSS with 25 mM HEPES and 0.5 mM EDTA) for 3 min, followed by buffer B (low-glucose DMEM with 0.4 mg/mL collagenase IV) for 5 min at a rate of 8–10 ml/min. After perfusion, the liver was transferred to a sterile Petri dish and gently shaken to disperse the hepatocytes. The isolated hepatocytes were filtered using a 70 μm strainer and purified by Percoll density gradient centrifugation. The purified hepatocytes were plated on collagen-coated dishes, incubated in

DMEM with 10% FBS for 4 h, and then maintained in DMEM without FBS until treatment.

Murine AML12 hepatocytes (ATCC) were maintained in a 1:1 mixture of DMEM and Ham's F12 medium, supplemented with 10% FBS, 1:100 ITS-G (Invitrogen), 100 U of penicillin/ml and 0.1 ng of streptomycin/ml. The AML12 cell line with a SHMT2-knockout was described previously[29]. The oxygen consumption rate was measured on a Seahorse XF analyzer (Agilent).

### Serum amino acid and blood biochemistry tests
The serum was prepared by centrifugation of the clotted blood at 10,000 g, 4 °C for 10 min. The serum amino acid levels were measured on an LC-MS/MS instrument (Waters) equipped with Cortec UPLC C18 column and Xevo TQ-S mass spectrometer using Kairos Amino Acid 100 Kit (Waters) as reference standards. Serum triglycerides and total cholesterol levels were measured using commercial kits from Wako and Pointe Scientific. Serum ALT was measured using the commercial kit from BioAssay Systems.

### Liver metabolite extraction and targeted metabolomics
For the isolation of hydrophilic metabolites, liver tissues were homogenized in 80% aqueous methanol by sonication. After centrifugation clarification, the supernatants were lyophilized on a SpeedVac vacuum concentrator. The metabolite extract pellets were stored at −80 °C until use. The metabolite levels were measured on an AB SCIEX LC-MS/MS system equipped with a SHIMADZU Nexera ultra high-performance chromatography system and a hybrid triple quadrupole and linear ion trap mass spectrometer, as described previously[40].

The metabolite abundance data were subjected to median normalization, log transformation, mean-center scaling, fold change analysis and two-sample *t*-tests using the MetaboAnalystR package available on the MetaboAnalyst 5.0 metabolomics analysis platform (https://www.metaboanalyst.ca/MetaboAnalyst/). The differentially expressed metabolites (DEMs) were identified using the significance cutoff of $p < 0.05$ and FDR < 0.20. The DEM pathway enrichment analysis was performed on MetaboAnalyst 5.0 using the Mus musculus SMPDB database. The results were visualized and plotted with GraphPad Prism 9.

### Liver lipid extraction and measurement
For lipid extraction, 50 mg of liver tissues were homogenized in 1 ml methanol by sonication on ice using a Branson Digital Sonifier. The homogenates were mixed with an additional 330 ul methanol and 2.66 ml chloroform by vertexing and incubated at 4 °C for 4 h, and then mixed with 1.6 ml 0.9% NaCl by vertexing followed by centrifugation at 4500 g, 4 C for 15 min. The organic phase was collected and evaporated under nitrogen flow. The lipid extracts were resuspended in isopropanol with 2%Triton. Liver triglycerides and total cholesterol levels were assessed using commercial kits obtained from Wako and Pointe Scientific, respectively.

### Glucose tolerance test
After a period of overnight fasting, mice were given an injection of glucose (2 g/kg body weight) into their peritoneal cavity. Blood samples were then taken from their tail veins at 0, 15, 30, 60, 90, and 120 min after the injection. These samples were tested for glucose concentration using a glucometer (Contour Next).

### Insulin tolerance test
After a 5-h period of fasting, insulin was administered to mice via an intraperitoneal injection at a dosage of 0.65U/kg of body weight. Blood samples were obtained from the tail vein at intervals of 0, 15, 30, 60, and 90 min post-insulin injection. The blood glucose levels were measured using a glucometer (Contour Next).

### Histological analysis
For H&E and Sirius Red staining, liver tissues were fixed in 10% neutral buffered formalin overnight, dehydrated into 70% ethanol, and then paraffin-embedded and sectioned. Tissue section slides were either stained

with hematoxylin and eosin (H&E, ThermoFisher Scientific); or stained with Picro-Sirius Red solution containing saturated aqueous picric acid with 0.1% Sirius red F3B (Sigma) for 1 h, followed by rinses with 0.5% acetic acid.

For Oil red O staining, formalin-fixed liver tissues were stepwise cryoprotected in 15% and 30% sucrose for 4 h and overnight, respectively, then embedded with OCT compound (Tissue-Tek) and flash-frozen in liquid nitrogen and stored at −80° C. The cryosections were fixed with 10% neutral buffered formalin for 30 min, rinsed with ddH₂O and 60% iso-propanol, and subsequently stained with 0.35% Oil Red O dye (Rowley Biochemical) for 15 min, followed by rinse in 60% isopropanol and counterstaining for nuclei with alum Hematoxylin.

The tissue section images were captured on a BZ-X800 microscope (Keyance) and scored using the following NAFLD histological scoring system: steatosis grade (0, <5%; 1, 5–33%; 2, >33–66%; 3, >66%), lobular inflammation (0, No foci; 1, <2 foci per 200x field; 2, 2–4 foci per 200x field; >4 foci per 200x field), and fibrosis stage (0, None; 1, perisinusoidal or periportal; 2, perisinusoidal and portal/periportal; 3, bridging fibrosis)[41].

## Western blotting

The tissues or cultured cell pellets were homogenized in RIPA lysis buffer (Tris-HCl pH7.4, 150 mM NaCl, 1% NP-40, 0.5% sodium deoxycholate, 0.1% SDS) by sonication on a Branson Digital Sonifier. Protein concentrations were determined with the BCA reagent (Pierce). Protein lysates (40 ug) were resolved by sodium dodecyl sulfate-polyacrylamide gel electrophoresis (SDS-PAGE), and proteins were transferred onto nitrocellulose filters. The blots were saturated with 5% milk and probed with antibodies against SHMT2 (#HPA020543, Sigma, 1:1000), NDUFS1 (sc-271510, Santa Cruz, 1:1000), Total OXPHOS Antibody Cocktail (ab110411, Abcam, 1:1000), ACLY (#15421-1-AP, Proteintech, 1:1000), ACC1 (#21923-1AP, Proreintech, 1:1000), FASN (#10624, Proteintech, 1:1000) or β-actin (A2066, Sigma, 1:1000). Following a wash with PBST (PBS containing 0.1% Tween 20), the blots were incubated with peroxidase-coupled goat anti-rabbit immunoglobulin G (Sigma, 1:5000). The immunolabeled protein bands were detected by enhanced chemiluminescence (ECL) method (Perkin Elmer) on an Azure C600 imager.

## RNA-seq, Bioinformatics, and qRT-PCR analysis

For RNA-seq transcriptomics profiling, total RNAs were isolated from liver tissues using RNeasy kit (Qiagen). The library construction and high throughput sequencing were conducted at Azenta Life Sciences. The cDNA libraries were constructed with the NEBNext Ultra RNA Library Prep Kit for Illumina (New England Biolabs), quantified with a Qubit 2.0 Fluorometer (Life Technologies) and a TapeStation (Agilent), and sequenced on a HiSeq 4000 sequencer (Illumina) using 2x150bp paired-end method. Base calling and FRSTQ file generation were achieved with the HiSeq Control and bcl2fasq 2.17 software. Sequence reads were trimmed to remove adapter sequences and poor-quality nucleotides using Trimmomatic v.0.36 and then mapped to the Mus musculus GRCm38 ERCC reference genome using the STAR aligner v.2.5.2b, resulting in the BAM files with >25 million total mapped reads per sample. The unique gene hits were extracted using EnsDb.Mmusculus.v79:Ensembl based annotation from R package, further normalized using edgeR and subjected to limma-voom-transformation and t-statistics calculation using the eBayes function of the limma R package to identify the differentially expressed genes (DEGs) with the significance cutoff of $p < 0.05$, adj $p < 0.25$. Gene set enrichment analysis (GSEA) were performed with the setting of m2.cp.wikipathways.v2023.1.Mm.symbols.gmt, 1000 gene_set-based permutations, chip platform = Mouse_Gene_Symbol_Remapping_MSigDB.v2023.1.Mm.chip, weighted enrichment statistic, and metric for ranking genes = tTest, and the significance cutoff of p,0.05 or FDR < 0.25.

For qRT-PCR analysis, total RNAs were isolated from liver tissues with the TRI Reagent (Sigma). cDNA libraries were constructed by random priming using the High-Capacity cDNA Reverse Transcription Kit (Applied Biosystems) and then used as a template for qPCR amplification with Luna Universal qPCR Master Mixes (New England Biolabs) on an

AriaMx cycler (Agilent). The mRNA levels were determined as the delta-delta threshold cycle ($\Delta\Delta C_T$) and normalized to the PPIA mRNA level. The PCR primers used are listed in Table S3.

## Human data analysis

The associations between SHMT2 common genetic variants and metabolic phenotypes in European Ancestry descendants were retrieved from the Common Metabolic Diseases Knowledge Portal (CMDKP, https://hugeamp.org/). The statistical analysis of the gene-level phenotypic associations was performed using the Multi-marker Analysis of GenoMic Annotation (MAGMA) method, available on the Portal platform, to calculate the bottom-line genetic associations.

For meta-analysis of *SHMT2* gene expression, the mRNA profiling for liver samples of healthy human donors ($n = 38$) and NASH patients ($N = 35$) were retrieved from the NCBI Gene Expression Omnibus under the accession of GSE89632[42] and GSE126848[43]. The unique gene counts were filtered, and subjected to TMM normalization and logCPM transformation with edgeR. The resulting SHMT2 expression values (log-transformed) were used to calculate the standardized mean difference (SMD, Hedge's g) between the groups of NASH and healthy control and presented as a forest plot using the random-effects model of the metafor R package.

## Statistics and reproducibility

All figures display individual data points representing single readings for each sample. Sample sizes and error management are specified in the figure legends.

## Reporting summary

Further information on research design is available in the Nature Portfolio Reporting Summary linked to this article.

## Data availability

The RNA sequencing data has been deposited in GEO under the accession number GSE248883. All other data supporting the results and conclusions of this paper are available in the paper and its Supplementary Information. The source data behind the graphs can be found in Supplementary Data 5. Uncropped images of gels can be found in Supplementary Fig. 5.

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

## Acknowledgements

We would like to thank Dr. James Granneman for the insightful discussion and critical reading of the manuscript. This study was supported by the National Institutes of Health R21AG064217 (J.W.), R21AG050741 (J.W.), and Barber Integrative Metabolism Program (J.W. and E.M.).

## Author contributions

J.W. and G.C. conceived and designed the study. G.C., L.Z., X.B., N.T., J.L., E.M., and J.W. performed the experiments and data analysis. G.Z. performed omics and GWAS data analysis. J.W. and G.C. wrote the paper.

## Competing interests

The authors declare no competing interests.
