## [Peer Review File · Communications Biology]

Reviewers' comments:

Reviewer #1 (Remarks to the Author):

this generally high quality manuscript explores the impact of hepatic SHMT loss on the development of fatty liver disease, finding evidence for increased liver fat accumulation but decreased fibrosis. Generally, the results are interesting and clearly presented, among the stronger and most enjoyable papers that I've reviewed lately.

Major suggestion:

1. Fig. 7 is very interesting but underdeveloped. I believe that SHMT1 is also linked to W-H ratio and the authors should include SHMT1 in their analysis of GWAS hits, as it catalyzes the same reaction.
2. Also, the authors should be more clear about the polymorphisms involved, their significance, and best guesses as to their biochemical and biological impact (e.g. by these coding or non-coding, how they impact SHMT2 transcript and protein levels, etc.).
3. Fig. 7B I guess is supposed to be showing SHMT2 transcript levels as a function of NASH from patient biopsies, but it's not clear enough what the GSE89632, SMD, etc. refer to or exactly how the data was handled and processed. It would also be great if the authors could get their hands on a few biopsies to check by IHC or similar.

Minor:

4. Fig 6: What are DEG?
5. The references to SHMT2 loss suppressing the mitochondrial electron transport chain should include the original articles in Nature and Mol Cell.
6. The Nature article on SHMT2 being essential for ETC expression is clear that the biochemical requirement is for mitochondrial meTHF. Hepatocytes are special in having alternative ways to acquire this (choline catabolism, glycine cleavage). This can be clarified for the readers to put the current results in context.
7. The paper should be more clear that it cannot distinguish between developmental and adult roles of hepatic SHMT2. These may differ in may impact therapy outcome with SHMT2 inhibitors.

Reviewer #2 (Remarks to the Author):

The study shows that SHMT2 in hepatocytes regulates 1C metabolism and inhibits the development of fatty liver disease. Interestingly, hepatic SHMT2-deficiency reduced liver inflammation and fibrosis in mice fed with an AMLN diet. This study described multiple phenotypes in hepatic SHMT2 knockout mice, potentially providing valuable insights into the function of hepatic SHMT2. However, the effects of SHMT2 on lipid metabolism, liver inflammation, and liver fibrosis have been previously reported, and the relationship among the multiple functions of SHMT2 remains unclear. Several concerns need to be addressed.

- 1) The level of liver and serum glycine is predominantly upregulated by knocking out the hepatic SHMT2. What role does the increase of glycine play?
- 2) Why does hepatic SHMT2 deficiency reduce liver inflammation and fibrosis in mice fed with an AMLN diet? Dose hepatic SHMT2 knockout affect liver injury?
- 3) Dose the increase of DNL by hepatic SHMT2 knockout affect liver inflammation and liver fibrosis?
- 4) In figure 1A, the western blot analysis of liver tissue samples is insufficient for verifying the conditional knockout of SHMT2 in hepatocytes.
- 5) In figure 5B, AML12 cells were used to investigate the role of SHMT2 in the mitochondrial respiration. Dose SHMT2 affect the mitochondrial respiration of primary mouse hepatocytes?

Responses to Reviewer #1's comments:

Reviewer #1 (Remarks to the Author):

this generally high quality manuscript explores the impact of hepatic SHMT loss on the development of fatty liver disease, finding evidence for increased liver fat accumulation but decreased fibrosis. Generally, the results are interesting and clearly presented, among the stronger and most enjoyable papers that I've reviewed lately.

Major suggestion:

1. Fig. 7 is very interesting but underdeveloped. I believe that SHMT1 is also linked to W-H ratio and the authors should include SHMT1 in their analysis of GWAS hits, as it catalyzes the same reaction.

Responses: We have carefully considered the Reviewer's suggestion and analyzed the SHMT1 data obtained from the Common Metabolic Disease Knowledge Portal (CMDKP). We have found that SHMT1 gene polymorphisms are significantly associated with the phenotypic variations of multiple critical human metabolic traits that define metabolic syndrome, including the W-H ratio. We have presented these findings in the new Fig. S3 and Table S6 and in the revised texts (Line 239-244). We agree that these observations significantly improve the understanding of the importance of the serine catabolic pathway in the pathogenesis of complex metabolic diseases.

2. Also, the authors should be more clear about the polymorphisms involved, their significance, and best guesses as to their biochemical and biological impact (e.g. by these coding or non-coding, how they impact SHMT2 transcript and protein levels, etc.).

Responses: We apologize for the lack of clarity. We have revised the text describing the recruitment criteria for SNPs to reflect their biological significance. (Line 230-235)

3. Fig. 7B I guess is supposed to be showing SHMT2 transcript levels as a function of NASH from patient biopsies, but it's not clear enough what the GSE89632, SMD, etc. refer to or exactly how the data was handled and processed. It would also be great if the authors could get their hands on a few biopsies to check by IHC or similar.

Responses: We apologize for the lack of clarity. We have revised the text to provide a clearer description of the parameters displayed in the figure. (Line 636-638)

We appreciate the suggestion made by the reviewer and are interested in investigating the status of SHMT2 protein in patient samples. Unfortunately, we do not have access to the required patient samples at the moment. We kindly request the reviewer to consider leaving this analysis for future studies.

Minor:

4. Fig 6: What are DEG?

Responses: We apologize for the lack of clarity. We have made a change to the text and spelled out the term "differentially expressed genes" instead of using the abbreviation "DEG". (Line 616)

5. The references to SHMT2 loss suppressing the mitochondrial electron transport chain should include the original articles in Nature and Mol Cell.

Responses: We have included the recommended citations in the revised version. Thanks. (Line 194)

6. The Nature article on SHMT2 being essential for ETC expression is clear that the biochemical requirement is for mitochondrial meTHF. Hepatocytes are special in having alternative ways to acquire this (choline catabolism, glycine cleavage). This can be clarified for the readers to put the current results in context.

Responses: We have revised the text to highlight the unique feature of hepatic 1C metabolism that differentially supports cellular respiration compared to other cell types. (Line 305-308)

7. The paper should be more clear that it cannot distinguish between developmental and adult roles of hepatic SHMT2. These may differ in may impact therapy outcome with SHMT2 inhibitors.

Responses: We have revised the text to emphasize the embryonic necessity of SHMT2 and the impact on the therapeutic implications of the pathway. (Line 364-366)

Responses to Reviewer #2's comments:

Reviewer #2 (Remarks to the Author):

The study shows that SHMT2 in hepatocytes regulates 1C metabolism and inhibits the development of fatty liver disease. Interestingly, hepatic SHMT2-deficiency reduced liver inflammation and fibrosis in mice fed with an AMLN diet. This study described multiple phenotypes in hepatic SHMT2 knockout mice, potentially providing valuable insights into the function of hepatic SHMT2. However, the effects of SHMT2 on lipid metabolism, liver inflammation, and liver fibrosis have been previously reported, and the relationship among the multiple functions of SHMT2 remains unclear. Several concerns need to be addressed.

1) The level of liver and serum glycine is predominantly upregulated by knocking out the hepatic SHMT2. What role does the increase of glycine play?

Responses: Glycine is a substrate that plays a crucial role in the production of glutathione, which is the primary antioxidant in the body. Previous studies have shown that glycine has a significant impact on the development of non-alcoholic fatty liver disease (NAFLD) by altering hepatic glutathione levels and antioxidant capacities (PMID 33268508). In our revision, we measured the levels of glutathione in the liver and found no significant difference in the levels when SHMT2 was deleted. We thus concluded that hepatic pathogenesis in SHMT2-deficient mice is most likely caused by 1-carbon deficiency rather than glycine-mediated oxidative stress. We have included this data in the new Figure S2 and discussed the findings in the revised texts. (Line 149-153; Line 310-316)

2) Why does hepatic SHMT2 deficiency reduce liver inflammation and fibrosis in mice fed with an AMLN diet? Dose hepatic SHMT2 knockout affect liver injury?

Responses: We conducted a study to measure the levels of alanine aminotransferase (ALT), which is an indicator of liver injury. Our findings suggest that deleting SHMT2 significantly

increases ALT levels in mice, regardless of whether they are fed chow or AMLN diets. Therefore, we have concluded that deleting SHMT2 promotes liver injury in mice. We have updated Figures 3 and 4 to include these data and discussed the findings in the revised texts. (Line 163-164; Line 166-167; Line 185-186)

We're still trying to fully understand why SHMT2-deficient mice experience reduced liver inflammation and fibrosis. Our metabolomic analysis suggests that a deficiency in 1C might alter the epigenetic landscape of hepatocytes, which could interfere with sensing hepatocyte injury, or it could disrupt nucleotide synthesis, which in turn prevents immune cell polarization and the activation of fibrogenic cells. We believe this is a fascinating issue that needs further exploration. We hope that the reviewer will allow us to leave this for future study.

3) Dose the increase of DNL by hepatic SHMT2 knockout affect liver inflammation and liver fibrosis?

Responses: We think that increased fat accumulation and damage in the liver resulted from increased lipogenesis. However, we do not think that this directly contributed to the reduced liver inflammation and fibrosis in the SHMT2 knockout mice. Typically, when the liver experiences fat accumulation and damage, it triggers inflammation and fibrosis. We believe that the deficiency in 1C, which plays a vital role in regulating gene expression and cell division, may be the underlying reason for the reduced immune response and activation of fibrosis in the SHMT2 knockout mice. We have discussed these findings in the revised texts. (Line 340-347)

4) In figure 1A, the western blot analysis of liver tissue samples is insufficient for verifying the conditional knockout of SHMT2 in hepatocytes.

Responses: We isolated primary hepatocytes from mice livers and confirmed the successful elimination of SHMT2 protein in the SHMT2 knockout mice hepatocytes using Western blotting. We have included this information in the revised Figure 5. Additionally, we have added a diagram in Figure S1 that illustrates the targeting strategy and genotyping outcomes of Cre/loxP recombination in the SHMT2 locus. We described the results in the revised texts. (Line 104-108; Line 196-198)

5) In figure 5B, AML12 cells were used to investigate the role of SHMT2 in the mitochondrial respiration. Dose SHMT2 affect the mitochondrial respiration of primary mouse hepatocytes?

Responses: We isolated primary hepatocytes from mouse livers and found that the deletion of SHMT2 did not affect their mitochondrial respiration. The results have been included in the revised Fig. 5. We discussed the results in the revised texts. (Line 196-202)

Figure 3 – Fig. 3g has been reformatted to Fig. 3h. A new Fig. 3g has been added to demonstrate ALT measurements.

Figure 4 – Fig. 4h has been added to demonstrate ALT measurements.

Figure 5 – Fig. 5b has been moved to Fig. S3. A new Fig. 5b has been added to demonstrate the data of primary hepatocytes.

Fig. S1 – A new Fig. S1 has been included to demonstrate the targeting strategy and verification for the desired recombination of the SHMT2 locus.

Fig. S2 – A new Fig. S2 has been added to demonstrate the measurements of glutathione.

Fig. S3 – This figure has been moved from Fig. 5 to demonstrate the data of AML12 cells.

Fig. S4 – The new Figure has been included to demonstrate the genetic association of the SHMT1 gene with human metabolic syndrome phenotypes.

REVIEWERS' COMMENTS:

Reviewer #2 (Remarks to the Author):

The authors have answered all of my questions and the paper has been greatly improved.